# Perspectives of Positively Charged Nanocrystals of Tedizolid Phosphate as a Topical Ocular Application in Rabbits

**DOI:** 10.3390/molecules27144619

**Published:** 2022-07-20

**Authors:** Abdullah Alshememry, Musaed Alkholief, Mohd Abul Kalam, Mohammad Raish, Raisuddin Ali, Sulaiman S. Alhudaithi, Muzaffar Iqbal, Aws Alshamsan

**Affiliations:** 1Department of Pharmaceutics, College of Pharmacy, King Saud University, Riyadh 11451, Saudi Arabia; aalshememry@ksu.edu.sa (A.A.); malkholief@ksu.edu.sa (M.A.); makalam@ksu.edu.sa (M.A.K.); mraish@ksu.edu.sa (M.R.); ramohammad@ksu.edu.sa (R.A.); salhudaithi@ksu.edu.sa (S.S.A.); 2Department of Pharmaceutical Chemistry, College of Pharmacy, King Saud University, Riyadh 11451, Saudi Arabia; muziqbal@ksu.edu.sa

**Keywords:** tedizolid, antimicrobial, nanocrystals, eyeirritation, ocular pharmacokinetics, transcorneal permeation

## Abstract

The aim of this study was the successful utilization of the positively charged nanocrystals (NCs) of Tedizolid Phosphate (TZP) (0.1% *w*/*v*) for topical ocular applications. TZP belongs to the 1, 3-oxazolidine-2-one class of antibiotics and has therapeutic potential for the treatment of many drug-resistant bacterial infections, including eye infections caused by MRSA, penicillin-resistant *Streptococcus pneumonia* and vancomycin-resistant *Enterococcus faecium*. However, its therapeutic usage is restricted due to its poor aqueous solubility and limited ocular availability. It is a prodrug and gets converted to Tedizolid (TDZ) by phosphatases in vivo. The sterilized NC_1_ was subjected to antimicrobial testing on Gram-positive bacteria. Ocular irritation and pharmacokinetics were performed in rabbits. Around a 1.29 to 1.53-fold increase in antibacterial activity was noted for NC_1_ against the *B. subtilis*, *S. pneumonia, S. aureus* and MRSA (SA-6538) as compared to the TZP-pure. The NC_1_-AqS was “practically non-irritating” to rabbit eyes. There was around a 1.67- and 1.43 fold increase in t_1/2_ (h) and C_max_ (ngmL^−1^) while there were 1.96-, 1.91-, 2.69- and 1.41-times increases in AUC_0–24h_,AUC_0–∞_,AUMC_0–∞_ and MRT_0–∞_, respectively, which were found by NC_1_ as compared to TZP-AqS in the ocular pharmacokinetic study. The clearance of TDZ was faster (11.43 mLh^−1^) from TZP-AqS as compared to NC_1_ (5.88 mLh^−1^). Relatively, an extended half-life (t_1/2_; 4.45 h) of TDZ and the prolonged ocular retention (MRT_0–∞_; 7.13 h) of NC_1_ was found, while a shorter half-life (t_1/2_; 2.66 h) of TDZ and MRT_0–∞_(t_1/2_; 5.05 h)was noted for TZP-AqS, respectively. Cationic TZP-NC_1_ could offer increased transcorneal permeation, which could mimic the improved ocular bioavailability of the drug in vivo. Conclusively, NC_1_ of TZP was identified as a promising substitute for the ocular delivery of TZP, with better performance as compared to its conventional AqS.

## 1. Introduction

Nanotechnology-based drug delivery systems (DDS) have overcome some of the pitfalls associated with conventional ophthalmic products (solutions, eye drops, suspensions, emulsions, etc.) such as improving the aqueous solubility and stability of poorly soluble/ lipophilic drugs [1,2,3]. In general, the frequent application of a topical ophthalmic dose (one–two drops) of any conventional eye drops of an antibiotic is needed in the affected eyes and only ~1–5% of the applied drug becomes available to the internal eye tissues. The poor ocular availability of conventional eye preparations have encouraged the development of novel nanocarriers-based ocular DDS, which would prolong the ocular retention of the applied dosage forms, permeate the drug(s) across the corneal and conjunctival area and improve the ocular (corneal and conjunctival) absorption and hence the bioavailability of the drug(s) together with minimizing eyeirritation/toxicity and visual interruption, as is associated with ocular gels [4,5].

In the present study, positively charged nanocrystals (NCs) of Tedizolid Phosphate (TZP) were used for ocular delivery. The NCs were prepared using asmall amount of stabilizer(s) with a drug [6,7,8], representing a good alternative to the existing colloidal nanocarriers such asnanoemulsions [9,10], microemulsion [11,12], liposomes [13,14], niosomes [15,16], polymeric nanoparticles (NPs) [17,18], dendrimer nanoparticles [19,20], solid lipid nanoparticles [3,21] and polymeric micelles [22,23,24], etc.

Despite some drawbacks associated with nanocarriers, their potential in ocular delivery for numerous drugs has been explored well, as these carrier systems have improved the ocular availability of many poorly soluble drugs while reducing the dosing frequency of the applied dose and hence any toxicity [17,23,25,26]. Moreover, the potential of NCs in ocular applications has remained relatively unnoticed due to the availability of numerous proven bioadhesive polymeric-NPs [26,27,28].

TZP is a phosphate monoester and a prodrug that gets converted to its active form Tedizolid (TDZ) by phosphatase enzymes during its in vivo fate [29,30]. TDZ is a 1,3-oxazolidin-2-one class of antibiotic, frequently used in the infections caused by drug-resistant bacteria, including the methicillin-resistant *Staphylococcus aureus* (MRSA), penicillin-resistant *Streptococcus pneumonia* and vancomycin-resistant *Enterococcus faecium*, etc. [31,32]. TDZ differs from the other members of 1, 3-oxazolidin-2-one by having a modified side chain at the C5 site of the1, 3-oxazolidin-2-one nucleus, which advises its action against some linezolid-resistant pathogenic microbes [33,34]. TDZ inhibits the bacterial protein synthesis by binding to the 23S rRNA of the 50S subunit of the ribosome, as is done by other oxazolidinone antibiotics [35]. The frequency of the occurrence of the resistance to TDZ is very low and it is 4–8-times more potent than linezolid against the species mentioned above [30]. The details of the structure–activity relationship (SAR) and the mechanism of action of TDZ have been explained well in our previous reports [26,36].

Due to the above reasons, we supposed that TZP might be a good choice of antibiotic in the present scenario of growing multidrug-resistant eye infections due to MRSA and many other resistant strains. In the present study, we investigated the in vitro antimicrobial efficacy of TZP-NCs against certain strains, the ocular irritation potential (if any) of NCs, the ocular pharmacokinetics of TDZ in rabbit eyes andthe ex vivo transcorneal permeation (through excised rabbit cornea) of TZP-NCs as compared to the conventional TZP-aqueous suspension (TZP-AqS). The developed TZP-NCs were characterized well andan in vitro release of TZP through the dialysis membrane was performed and reported in the previous part of this article [37]. The previously reported LC-MS/MS method was successfully utilized for the quantitative determination of TDZ in rabbit aqueous humor samples.

## 2. Materials and Methods

### 2.1. Materials

Tedizolid and Tedizolid Phosphate (C_17_H_15_FN_6_O_6_P; MW 450.32 Da) with more than 98% purity were purchased from “Beijing Mesochem Technology Co., Ltd. (Beijing, China)”. Ketamine. HCl(TEKAM^®^, 50 mgmL^−1^) was purchased from HIKMA Pharmaceuticals (Amman, Jordan). Stearylamine and mannitol were purchased from Alpha Chemika, Mumbai, India and Qualikems Fine Chem Pvt. Ltd. (Vadodara, India), respectively. The HPLC grade methanol and acetonitrile were purchased from “BDH Ltd. (Poole, England)”. Polyvinyl alcohol (Mw 16,000), Poloxamer-188 (Pluronic-F68), Sodium Lauryl Sulfate and Benzalkonium chloride were purchased from Sigma Aldrich (St. Louis, MO, USA). Milli-Q^®^ water was obtained by a Millipore filter unit (Millipore, Molsheim, France). All the other chemicals and solvents were of analytical grade and HPLC grade, respectively.

### 2.2. Methods

#### 2.2.1. Nanocrystals of Tedizolid Phosphate

The NCs of TZP were formulated by the antisolvent precipitation technique, using homogenization and probe sonication steps. The optimal formulation (TZP-NC_1_) was well characterized. The characterization parameters included the size, polydispersityindex, zetapotential, structural morphology by scanning electron microscopy, FTIR for any interaction with the excipients, crystallinity by differential scanning calorimetry and X-ray diffraction studies, physicochemical characterization of the NCs for ocular suitability, saturation solubility, in vitro drug release in simulated tear fluid and storage stability at three different temperatures for 6 months. The data regarding these experiments have been published as a separate article in another journal [37]. For the ease of the reader, here we have included the composition of the formulations as mentioned in Table 1. Therefore, here only was the optimized formulation further subjected to the following studies for its in vivo ocular suitability in rabbits.

#### 2.2.2. Sterilization and Sterility Testing

The final formulations (aqueous suspensions of TZP-NC1 and TZP-pure) were prepared in an aseptic area as per the guidelines available concerning the aseptic filling method for the ophthalmic dosage forms because aseptic processing is highly regulated with considerable guidance in the US Code of Federal Regulations (CFR 21), FDA documents and the EU-GMPS “Rules and Guidance for Pharmaceutical Manufacturers and Distributors” [38,39]. Although the final products were prepared in an aseptic area, we still performed the sterilization of the products because these were intended for in vivo studies in rabbit eyes. Terminal sterilization by autoclaving in the final container is possible for the products if the stability of the drugs/products is not adversely affected by the moist heat (121 °C). Considering these facts, therefore, the AqS of TZP-NC_1_ and TZP-pure were aseptically filled in the HDPE container. Before the aseptic filling, the bulk preparation TZP-NC1 was sterilized by filtration. TZP-NC1 was filtered through a 0.22 μm membrane filter into the final sterile 10 mL capacity HDPE container. Such membrane filters can remove most of the bioburden including bacteria and fungi [40]. TZP-AqS was not terminally sterilized; rather, it was prepared in the aseptic area using freshly autoclaved Milli-Q water.

The sterility testing of the sterilized TZP-NC_1_ was performed according to the USP method [41]. Briefly, two containers of TZP-NC_1_ were tested for sterility. TheTZP-NC_1_-AqS (2 mL) from the two containers of sterilized products was pooled out in the aseptic condition. The pooled samples were further diluted with 8 mL of autoclaved double distilled water. The sterile syringe filter (0.22 μm pore size) is a type of membrane filter (Corning Inc., New York, NY 14831, USA) that was fixed in a membrane-filter funnel unit. The filter was moistened with Fluid-A (1 g of peptic digest of animal tissues in 1000 mL of distilled water). The diluted pooled TZP-NC_1_ suspension was then passed through the membrane filter in an aseptic condition. As the product contained an antimicrobial agent (TZP), the membrane was washed repeatedly (4 times) with 100 mL of sterilized Fluid-A. Thereafter, the membrane was then divided into two parts; one part was transferred to Soybean Casein Digest Media (for molds/fungi and lower bacteria) and was incubated at 20–25 °C for 10 days, and the other portion of the membrane was put into Fluid Thioglycollate Media (for aerobic and/or anaerobic bacteria) and was incubated at 30–35 °C for 10 days.

#### 2.2.3. Antimicrobial Study

The antimicrobial activity of the TZP-NCs-AqS and conventional TZP-AqS was accomplished through the agar diffusion method [26,42]. The bacterial strains for this testing were chosen from the “Global Priority Pathogens List” and are available at “Department of Pharmaceutics, College of Pharmacy, King Saud University”. A total of four Gram-positive American Type Culture Collections (ATCC) of *Bacillus subtilis*, *Streptococcus pneumonia, Staphylococcus aureus* and MRSA (SA-6538) were used for their susceptibility toward TZP. The MHA plates were aseptically prepared and the chosen strains were spread out on the separate MHA-containing plates. Using a sterile borer, three wells of ~6 mm diameter were made. In the first well, 30 µL of conventional TZP-AqS (30.0 µg of TZP) was inoculated, in the second well, 30 µL of TZP-NC_1_-AqS (30.0 µg of TZP)was inoculated, and in the third well, the same volume of blank AqS without TZP was transferred. All the plates were left untouched for 1 h for the proper diffusion of the products into the medium and the plates were incubated at 37 °C for 24 h. Thereafter, the zones of inhibition created by the test products on the plates were measured. The antimicrobial assessment was accomplished in triplicate. The results are represented as the mean ± SD of the three measurements. Statistical analysis was performed using GraphPad Prism: Version 5 (GraphPad Software, Inc., San Diego, CA, USA). A oneway analysis of variance followed by Tukey’s multiple comparison test was conducted by considering *p* < 0.05 as statistically significant.

#### 2.2.4. In Vivo Animal Study

New Zealand white rabbits weighing 2.0–3.0 kg were used for thein vivo studies. The protocol for animal use was approved by the Research Ethics Committee at King Saud University (approval No. KSU-SE-18-25, amended). The animals were housed in air-conditioned rooms with 75 ± 5% relative humidity, as per the “Guide for the Care and Use of Laboratory Animals”. All the animals were healthy (free from ocular problems). “The animals were kept on a standard pellet diet and watere *ad libitum* and ”fasted overnight before starting the experiment.

##### Eye Irritation

This study was performed on healthy rabbits by following Draize’s test [43]. We followed the guidelines of “The Association for Research in Vision and Ophthalmology (ARVO)” for animal use in “Ophthalmic and Vision Research”. So, only the left eyes of the animals were selected for the test samples and the right eyes were left untreated. Based on the characterizations to obtainan optimized formulation, the nanocrystals (NC_1_) were considered for eye irritation tests as compared to conventional TZP-AqS.

Generally, six rabbits are taken for one test product; in the present investigation, we used three animals for one test product, as there might have been a chance of severe ocular irritation and damage [44]. Additionally, we had constraints with the number of animals used. Six rabbits were divided into two groups, three for NC_1_ and three for TZP-AqS (conventional). For acute irritation, three consecutive doses (at 10 min intervals) of TZP-AqS and the suspension of NC_1_ (40 μL) were instilled in the right eyes of each animal of the respective groups. After one hour of dosing, the eyes were visually observed periodically for 24 h for any injuries or signs and symptoms in the conjunctiva, iris and cornea or for any changes in the treated eyes other than that of the NaCl treated. The photographs of the eyes were clicked for scoring. Additionally, the level of irritation was assessed [45] based on the discomfort to the animals and signs and symptoms including redness, swelling (edema), chemosis in the conjunctiva, cornea and iris, or mucoidal/non-mucoidal discharge [28]. The scoring was performed and the irritation (if any) due to NC_1_ was characterized as per the designated system [46,47].

##### Ocular Pharmacokinetics (PK)

The TDZ concentration in the aqueous humor (AqH) was determined to check the ocular bioavailability of the active form of the drug (active form) after the topical ocular application of the TZP (prodrug)-containing formulations in the healthy rabbits. Six rabbits were divided into two groups (one for TZP-NC_1_ and the second for TZP-AqS). Forty microliters (40 μL, equivalent to 40 μg of TZP) of the sterilized formulations were applied to the left eyes of the rabbits of the respective groups [17,28]. Half an hour post dosing, the rabbits were desensitized with an intravenous injection of a Ketamine. HCl and Xylazine mixture [17,28,48]. Subsequently, around 40 µL of the AqH was aspirated by a 29-gauge needle attached to an insulin syringe at stipulated times. The collected samples were prepared and analyzed by liquid chromatography and the mass spectrometric (LC-MS/MS) method [36].

##### Chromatography of TDZ and Mass Spectrometric Conditions (LC-MS/MS)

The chromatographic and mass spectrometric conditions for the analysis of TDZ were previously reported by our group in detail [36]. Briefly, the “UPLC system (Acquity™) connected with a triple-quadruple Tandem Mass-Spectrometer Detector (TQD) (Waters^®^, Milford, MA, USA)” was used. The chromatographic separation of TDZ (active moiety) and Linezolid as the internal standard (IS) was accomplished on “Acquity™ HILIC column (2.1 × 100 mm, 1.7 μm)”, fitted with 0.22 μ of a stainless-steel fritfilter (Waters^®^, Milford, MA, USA). The column temperature was maintained at 40 °C. The mobile phase was composed of acetonitrile and 20 mM of ammonium acetate at an 85:15 (*v*/*v*) ratio and was pumped at a 0.3 mLmin^−1^ flow rate. The injection volume was 3 μL and the total runtime was 3 min for the elution of the drugs. The Tedizolid (TDZ) and the IS were eluted with retention times (*R*_t_) of 1.12 and 1.32 min, respectively. The TQD fitted with the electrospray ionization interface was operated in positive mode for the detection of the two elutes. The optimal “TQD parameters were: the source temperature (150 °C), capillary voltage (3.7 kV), dwell time (0.161 s), desolvation temperature (350 °C), desolvation gas (N_2_) flow rate (600 L.h^−1^), cone gas flow rate (50 L.h^−1^) and collision gas (Argon) flow rate (0.13 mL.min^−1^)”. The optimal MS/ MS conditions including the cone voltages were 32 V and 34 V (for TDZ and IS, respectively) whereas the collision energy was 18 eV (for both elutes). The “Multiple reactions monitoring (MRM) was used for the quantification of TDZ and IS with the parent to daughter ion transitions (*m*/*z*) of 371.15→343.17 and 338.18→296.22, respectively”. “The UPLC-MS/MS system was operated by Mass-Lynx Software (V-4.1, SCN-714)” while the obtained chromatograms were processed by the “Target Lynx^TM^ program” as reported [36,49].

#### 2.2.5. Transcorneal Permeation

The transcorneal permeation of TZP from NC_1_ across the excised rabbit cornea was performed using “fabricated double-jacketed transdermal diffusion cells assembled with the automated sampling system-SFDC 6, LOGAN, Somerset, NJ, USA”(a schematic representation of the Franz diffusion cell is shown in Appendix A, appeared in Appendix A) [28]. After three weeks (the washout period) of the irritation study, the same rabbits were sacrificed by an overdose intravenous injection of a Ketamine. HCl and Xylazine mixture (15 and 3 mgkg^−1^ b. wt., respectively). The left eyes (used as the control in Draize’s test) were taken out and the corneas were excised and fitted between the donor and receptor components of the diffusion cells, where the epithelial layer of the cornea was towards the donor component. The STF with SLS (0.5% *w*/*v*) was filled in the receptor component and a small magnetic bead was also put into it. The filled cells were placed on different stations of the LOGAN instrument and water (at 37 ± 1 °C) was allowed to flow into the outer jacket of the cells. For each formulation (in triplicate), 500 μL (0.1%, *w*/*v*) of the suspension of NC_1_ and the drug-aqueous suspension (TZP-AqS) was put into the donor components and the instrument was switched on with magnetic stirring. Sampling was conducted from the receptor component at different time points until 4 h and the drug (μgmL^−1^) was analyzed by the HPLC-UV method as mentioned above. The permeated amount of the drug (μgcm^−2^) through the cornea was calculated. The calculation was performed by considering the volume of the receptor compartment (5.2 mL), where DF stands for the dilution factor, as well as the involved corneal cross-section area (0.5024 cm^2^) and the initial drug concentration (1000 µgmL^−1^), using Equation(1).
(1)$$\mathrm{Permeated}\;\mathrm{mount}\;\mathrm{of}\;\mathrm{drug}\;\left({µ\mathrm{gcm}}^{-2}\right)=\frac{\mathrm{Conc}.\;\left({µ\mathrm{gmL}}^{-1}\right)\times \mathrm{DF}\times \mathrm{Volume}\;\mathrm{of}\;\mathrm{receptor}\;\mathrm{compartment}\;\left(\mathrm{mL}\right)}{\mathrm{Area}\;\mathrm{of}\;\mathrm{cornea}\;\mathrm{involved}\;({\mathrm{cm}}^{2})}$$

The slope of the time versus permeated amount plot was applied to determine the permeation parameters (flux and apparent permeability/permeability coefficient) using the following Equations (Equations (2) and (3)):(2)$$\mathrm{Steady}\;\mathrm{state}\;\mathrm{flux}\;i.e.,\;J\;\left(µgcm^{-2}.h^{-1}\right)=\frac{dQ}{dt}\;$$
(3)$$\mathrm{Apparent}\;\mathrm{permeability}\;i.e.,\;P_{app}\;\left(cm.h^{-1}\right)=\frac{J}{C_{0}}$$
where “*Q*” = amount of drug passed through the excised cornea, (*^dQ^/_dt_*) = linear ascent of the slope, “*t*” = contact time of formulation with corneal epithelial layer and “*C*_0_” = initial concentration of TZP.

Moreover, after finishing the transcorneal permeation studies, the used corneas were weighed, dipped into 1.0 mL of methanol, left overnight to be dried at around 75–80 °C and then reweighed. From the weight differences, the corneal hydration level was estimated [3,50].

#### 2.2.6. Statistical Analysis

The results are represented as the mean with standard deviation (±SD) unless otherwise indicated (as ± SEM was used for the PK parameters). Statistical analysis was performed using GraphPad Prism: Version5 (GraphPad Software, Inc., San Diego, CA, USA). A non-compartmental approach was used for the estimation of the PK parameters by “*PK*-Solver Software, Nanjing, China using MS-Excel-2013” [51]. The comparative analysis of the data was accomplished by the Student’s *t*-test and *p* < 0.05 was considered statistically significant.

## 3. Results and Discussion

### 3.1. Formulation and Characterization of the Optimized Formulation

The optimized nanocrystal (TZP-NC_1_) was suitable for ocular use, having a size range of 154.3 ± 17.9 nm with good crystalline morphology, a good polydispersityindex (0.243 ± 0.009) and a zetapotential of +31.6 ± 3.8 mV. The smaller particle size and larger surface area of the nanocrystals helps them to cross the mucus layer of the tear film, which increases the residence time of formulation in the eye by keeping them in contact with the corneal tissues. The increased contact time with cornea may increase the absorption, which further translates into an improved bioavailability. The nanocrystals are also responsible for the increased corneal permeation, which will help in the treatment of intraocular diseases. The positive zeta potential values suggest an enhanced electrostatic interaction of the nanocrystals with the negatively charged mucin layer, which helps with increasing the residence time of the drug in the eye. The freeze drying of NC_1_ with mannitol (1%, *w*/*v*) provided good stabilization to NC_1_, prevented crystal growth and provided iso-osmolarity to the NC_1_-suspension after redispersion in dextrose (5%, *w*/*v*), where the drug content was 96.4%. The FTIR spectroscopy indicated no alteration in the basic molecular structure of TZP after nanocrystallization, and the DSC and X-ray diffraction validated the reduced crystallinity of TZP-NC. The solubility of NC_1_ in the simulated tear fluid (STF) with sodium lauryl sulfate (SLS, 0.5%, *w*/*v*) resulted in a 1.6-fold increase as compared to the pure TZP due to its nanosizing. The redispersion of freeze-dried NC_1_ produced a clear transparent aqueous suspension of NC_1_ with osmolarity (≈298 mOsm.L^−1^) and viscosity (≈21.07 cps at 35 °C). A relatively higher (≈78.8%) release of the drug from NC_1_ was obtained as compared to the conventional TZP-aqueous suspension (≈43.4%) at 12 h in STF with SLS (0.5%, *w*/*v*). The NC_1_ was found to be physically (size, PDI, ZP) and chemically (drug content) stable at 4 °C, 25 °C and 40 °C for 6 months. The above findings encourages us that the topical ocular application of TZP-NC_1_ in rabbits is one of the best alternatives to the conventional aqueous suspension of the poorly soluble TZP with an improved performance.

### 3.2. Interpretation of Sterility Testing

During the incubation, both media were visually observed every day for 10 days to see the appearance of any turbidity due to microbial growth. The media should be clear and transparent against a light source. The appearance of turbidity or cloudiness in the media is indicative of microbial growth. In the present study, no turbidity/cloudiness was found in any of the two culture media. Thus, the tested sample of TZP-NC_1_-AqS passed the sterility test and could be suitable for ophthalmic purposes.

### 3.3. Antimicrobial Activity

The results of theantimicrobial activity experiment using the “agar diffusion method” aresummarized in Table 2. The TZP-NC_1_ showeda significant improvement (*p* < 0.05) in the antimicrobial action against all the tested Gram-positive microbes (*Bacillus subtilis*, *Streptococcus pneumonia, Staphylococcus aureus* and MRSA (SA-6538)) as compared to the conventional TZP-AqS, as illustrated in Figure 1. Relatively very little antimicrobial activity was illustrious for the blank-AqS than those of the tested TZP-containing products. Such little activity by the blank-AqS was due to the presence of some antibacterial excipients (those added in the AqS except TZP), such as the quaternary ammonium benzalkonium chloride (0.01%) which has broad-spectrum antibacterial activity and acts by interacting with the negatively charged bacterial membrane [52] and polyvinyl alcohol [53]. The antibacterial activities in the present investigation were further substantiated by the previous findings, where an improved activity of TZP-loaded chitosan nanoparticles was reported against the conventional TZP-AqS [26].

The level of significance between the two TZP preparations in comparison to the blank AqS against the used microbes was performed using a one-way analysis of variance (a one-way ANOVA) followed by Tukey’s multiple comparison test using GraphPad Prism V-5.0 by considering the *p* < 0.05 as statistically significant; the data obtained arerepresented (Table 1).The antimicrobial activity of TZP-NC_1_ was enhanced as compared to the conventional TZP-AqS. These findings also pointed out that the process of nanocrystallization could not adversely affect the fundamental (antimicrobial) property as well as the structure–activity relationship (SAR) of the oxazolidinone antibiotic (TZP) in the present investigation. Both of the TZP preparations showed significantly (*p* < 0.05) increased antimicrobial activity as compared to the blank AqS (no activity).Thus, we could assume that the size reduction following the nanocrystallization increased the antimicrobial effectiveness of TZP. This might be attributed to the fact that the size reduction in the crystals could increase the aqueous solubility of the highly lipophilic drug (TZP), which increased the drug intake into the bacteria and inhibited their protein synthesis by binding to the 23S rRNA of the 50S subunit of the ribosome [35,54].

### 3.4. Eye Irritation

The ocular irritation (if any) caused by the application of NC_1_ as compared to TZP-AqS (conventional) was investigated for 24 h by considering the NaCl-treated eyes as normal. Any alterations in the cornea, conjunctiva and iris were visually observed [55]. Based on the signs and symptoms of eye irritation, the scoring for irritation was performed by following the grading and scoring systems (Appendix A, appeared in Appendix A). The signs and symptoms included redness, swelling, hemorrhage, chemosis, cloudiness (mucoidal) and edema, etc., which could have possibly occurred in the treated eyes [46]. The type of irritation was categorized according to the ocular irritation classification [47] mentioned in Appendix A (appeared in Appendix A). The obtained scores during the experiment for the test samples are summarized in Table 3.

No clear signs of ocular discomfort were noted in the treated rabbits during the irritation testing of NC_1_ as compared to TZP-AqS. Figure 2a,a’ are the representative images of the normal saline (NaCl, 0.9%)-treated eyes for the TZP-AqS- and NC_1_-treated animals, respectively. Figure 2b,b’ show the redness of the conjunctiva with mild mucoidal discharge (red arrow) after 1 h post application of TZP-AqS and NC_1_, respectively. Among the three rabbits treated with TZP-AqS, one showed mild redness (less intense) and mucoidal discharge even at 3 h (Figure 2c, red arrow), while the NC_1_-treated rabbits did not show any such abnormal ocular discharge at 3 h (Figure 2c’, green arrow).The less intense redness and mucoidal discharge by the TZP-AqS-treated rabbits at 3 h was probably due to the larger size of the suspended particles and PVA (which was added as a suspending agent in AqS), which caused some corneal abrasion. Hence, to overcome such unwanted phenomena, the ocular physiological secretions (mucoidal discharge) occurred and such secretions remained until 6 h (Figure 2d, black arrow), while no such signs and symptoms were noted at 6 h in the eyes of the NC_1_-treated rabbits. The redness in the treated eye was much reduced or almost recovered and clear, as denoted by the green arrow (Figure 2d’). The redness of the conjunctiva and ocular inflammation completely disappeared from their normal state (green arrows) at 24 h post topical application of TZP-AqS and NC_1_, as illustrated in Figure 2e and 2e’, respectively. The redness of the conjunctiva and eye inflammation was gone and the eyes regained their normal conditions after 24 h postapplication of the test products. The disappearance of such symptoms was due to the natural defense system of the eyes and the use of the Generally Recognized as Safe (GRAS) excipients in the formulations [22,26].

The ocular application of TZP-AqS caused minimal irritation in one rabbit with redness of the eye and mucoidal discharge, which was given a score of one. No corneal lesions or opacity were observed; hence, the cornea, conjunctiva and iris were given a score of 0 (Table 3). A reported classification system for irritation scoring [47] was followed to calculate the maximum mean total score (MMTS). The MMTS for TZP-AqS and TZP- NC_1_-AqS after 24 h of their application was 8.00 (>2.6 and <15.1, minimally) and 1.33 (>0.6 and <2.6, practically none), as mentioned in Table 4. Thus, the conventional TZP-AqS was “minimally irritating”, while the TZP-NC_1_-AqS was “practically non-irritating” to the rabbit eyes; thus, there is hope for its ocular application. All the involved animals remained healthy and active without any odd signs of ocular irritation during the experiment, except a few as stated above. Thus, we concluded that the conventional TZP-AqS, as well as the developed TZP-NC1-AqS, were well tolerated by the rabbit eyes.

### 3.5. Ocular Pharmacokinetics

The previously developed LC-MS/MS method by our group was effectively used for the quantification of TDZ in the aqueous humor (AqH) obtained from the rabbit eyes [36]. The level of TDZ in the AqH versus time plots and the calculated pharmacokinetic parameters for the two TZP formulations are, respectively, illustrated in Figure 3 and Table 5. After the topical application of the two formulations, a fast release of the drug (TZP) was found from NC_1_-AqS as compared to the conventional TZP-AqS, indicating a faster absorption of TDZ and an attained a maximum concentration (C_max_) of 829.21 ± 38.27 ngmL^−1^ and 580.92 ± 45.48 ngmL^−1^, respectively, at 2 h of T_max_. Thereafter, the concentration of the active form of the drug in the AqH decreased in a log-linear fashion, with the average elimination half-lives of 4.45 h and 2.66 h for NC_1_ and TZP-AqS, respectively, just after the second sampling at 2 h, which was indicative of the fast absorption (up to 2 h) of TDZ from NC_1_ as compared to its counter formulation.

Other than T_max_, the differences were statistically significant (*p* < 0.05) in the rest of the pharmacokinetic parameters for the two formulations. Around a 1.67- and 1.43-fold increase in t_1/2_ (h) and C_max_ (ngmL^−1^), respectively, was found from NC_1_ as compared to the pure drug suspension (TZP-AqS). Approximately 1.96-, 1.91-, 2.69- and 1.41-times increases in AUC_0–24h_, AUC_0–∞_, AUMC_0–∞_ and MRT_0–∞_, respectively, were obtained for the active form of TZP from NC_1_ than that of TZP-AqS. The clearance (CL/F) of TDZ was faster from TZP-AqS (11.43 mLh^−1^) than that of NC_1_ (5.88 mLh^−1^). The faster clearance of the drug from TZP-AqS could be the primary reason for the relatively low ocular bioavailability, which was further justified by the extended half-life (t_1/2_; 4.45 h) of TDZ and the prolonged ocular retention (MRT_0–∞_; 7.13 h) of NC_1_ compared to the shorter half-life (t_1/2_; 2.66) of TDZ and MRT_0–∞_ and of TZP-AqS (t_1/2_; 5.05 h), which was further confirmed by the fast elimination of TDZ as the drug concentration was not detectable at 24 h from AqS.

Overall, the comparative pharmacokinetic profiling illustrated an improved ocular bioavailability of TDZ from NC_1_ as compared to TZP-AqS. This might be due to the high positive ZP (+29.4 mV) of NC_1_ (due to the presence of Benzalkonium chloride and stearylamine), which could interact electrostatically with the negatively charged mucin layer on ocular surfaces and increase the contact time of NC_1_. This interaction could help improve the penetration of NC_1_ across the cornea, which in turn could improve its cellular uptake and hence the bioavailability of TDZ [6,26,36]. Additionally, the nanosize range of NC_1_ could also be a reason for its increased transcorneal permeation. Similarly, the significantly increased ocular bioavailability of hydrocortisone and some other poorly soluble glucocorticoids were reported from the nanosuspension more than their micro-range formulations [56]. Conclusively, the nanocrystallization of TZP could have the potential to enhance the ocular bioavailability of TDZ at a relatively low dose and could have the reduced dosing frequency of the TZP-NC_1_-AqS as an ophthalmic formulation.

### 3.6. Transcorneal Permeation

The cumulative amounts of the drug that were permeated (µgcm^−2^) were plotted against time (h), as shown in Figure 4, and from these plots the permeation parameters were calculated and summarized in Table 6. The TZP-NC_1_-AqS showed a linear permeation of TZP as compared to the conventional TZP-AqS up to 4 h. Overall, the cumulative amounts of the permeated drug were 44.32 ± 1.74 and 70.43 ± 3.52 µgcm^−2^(at 4 h) from the conventional TZP-AqSand TZP-NC_1_-AqS, respectively.

The transcorneal permeation of TZP was higher from AqS-NC_1_ throughout the experiment as compared to its counter formulation. In the case of NC_1_, around 34.94 ± 3.58 µgcm^−2^ of TZP was traversed at 1.5 h, while roughly around same amount of the drug (35.92 ± 1.94 µgcm^−2^) was passed through the cornea at 2 h from the conventional TZP-AqS. Around 42.1 µgcm^−2^ of the drug was crossed at 2.5 h from the TZP-AqS; after that, the amount of crossed drug through the cornea was not increased significantly (*p* < 0.05) and it was comparable (44.32 µgcm^−2^) until 4 h. However, an increased permeation of TZP (51.15 µgcm^−2^) was noted from the NC_1_ at 2.5 h and there was a linear progression until 4 h (70.42 µgcm^−2^). Overall, the permeated amount of drug from the NC_1_ was significantly (*p* < 0.05) increased as compared to the TZP-AqS. The enhanced permeation of TZP from the NC_1_ form indicated that the nanocrystallization of the drug improved its solubility in the aqueous media. The reduced size of the TZP-NC_1_ could easily cross the cornea as compared to the particle size (574.5 nm) of TZP-AqS. The pH of the formulations (Table 6) were appropriate for the transcorneal permeation of TZP. Additionally, the partitioning of a neutral drug species between *n*-octanol/water (LogP) of TZP is around 4.89 at a neutral pH (7.0). Therefore, it was assumed that, as the pH of the TZP-NC_1_-AqS was closer to the neutral pH or the pH of the tear fluids, a larger fraction of the TZP remained unionized in the NCs of TZP as compared to its conventional AqS [57], which might be attributed to the enhanced transcorneal permeation of TZP from NC_1_. Conclusively, from the permeation profiles, the NC_1_ form of TZP could offer a linearly increased permeation of the drug, which could mimic the improved ocular bioavailability of the drug during in vivo application of TZP-NC_1_-AqS as compared to its counter formulation (conventional TZP-AqS).

The corneal hydration levels were found to be 77.29 ± 1.23% and 78.05 ± 1.27% for the conventional TZP−AqS- and TZP−NCs−AqS-treated excised rabbit corneas, respectively. The obtained values were in the range of a normal hydration level (between 75% to 80%) [58] at pH 6.18 (for conventional TZP-AqS) and 7.03 (TZP-NCs−AqS) as summarized in Table 6. The corneal hydration levels as mentioned in Table 6 were below 80%; therefore, the damage that appeared during the ex vivo transcorneal permeation experiment to the corneal was considered reversible and non-damaging [58].

## 4. Conclusions

Around a 1.29 to 1.53-fold increase in antibacterial activity was noted against *B. subtilis*, *S. pneumonia, S. aureus* and MRSA (SA-6538) as compared to the pure TZP. The ocular irritation study indicated that the conventional TZP−AqS was “minimally irritating” and NC_1_-AqS was “practically non-irritating” to the rabbit eyes; thus, there is hope for its ocular application. Around a 1.67- and 1.43-fold increase in t_1/2_ (h) and C_max_ (ngmL^−1^) occurred, while 1.96-, 1.91-, 2.69- and 1.41-times increases in AUC_0–24h_, AUC_0–∞_, AUMC_0–∞_ and MRT_0–∞_, respectively, were found for TDZ (active of TZP) by NC_1_ as compared to TZP−AqS. The clearance of TDZ was slower (5.88 mLh^−1^) from NC_1_ as compared to TZP−AqS (11.43 mLh^−1^).This was further substantiated by the extended half-life (t_1/2_; 4.45 h) of TDZ and the prolonged ocular retention (MRT_0–∞_; 7.13 h) of NC_1_ as compared to the shorter half-life (t_1/2_; 2.66) of TDZ and MRT_0–∞_, as well as of TZP−AqS(t_1/2_; 5.05 h), and due to fast elimination rate of the conventional AqS, the concentration of TDZ was not detected in the 24 h AqH samples. The cationic TZP−NC_1_ could offer an increased transcorneal permeation of the drug, which could mimic the improved ocular bioavailability of the drug in vivo. Summarily, the cationic NC_1_ of TZP is a promising alternative for the ocular delivery of TZP, with an amplified performance comparatively to the conventional TZP−AqS. Further, in vivo studies (rabbit uveitis models) are warned to check the anti-inflammatory activity of the drug during bacterial eye infections, including the different ocular anterior and posterior segment inflammatory conditions and some retinal ailments following the topical application of the developed TZP−NC_1_.

## Figures and Tables

**Figure 1 molecules-27-04619-f001:**
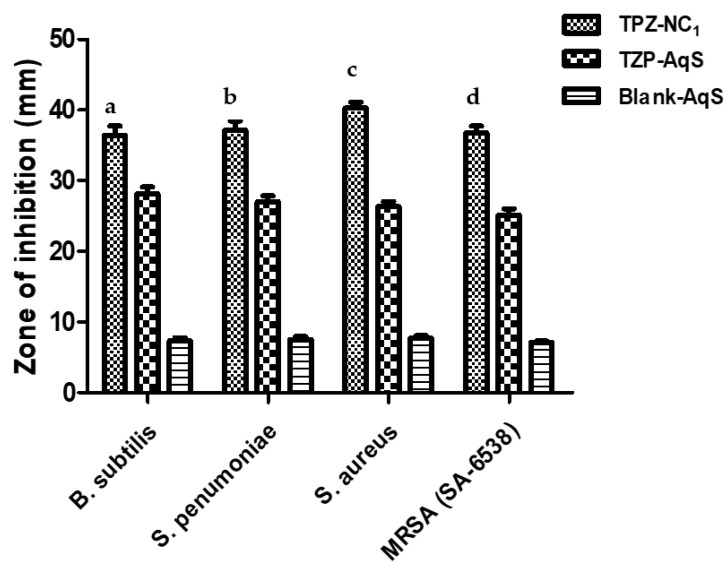
Antimicrobial activity of TZP-containing products as compared to the blank aqueous suspension (AqS) against *Bacillus subtilis*, *Streptococcus pneumonia*, *Staphylococcus aureus* and MRSA (SA-6538). Results are represented as mean with SD of three measurements. “a” *p* < 0.05, TZP-NC_1_ versus other test substances (for *B.subtilis*); “b” *p* < 0.05, TZP-NC_1_ versus other test substances (for *S. pneumonia*); “c” *p* < 0.05, TZP-NC_1_ versus other test substances (for *S.aureus*); “d” *p* < 0.05, TZP-NC_1_ versus other test substances (for MRSA SA-6538).

**Figure 2 molecules-27-04619-f002:**
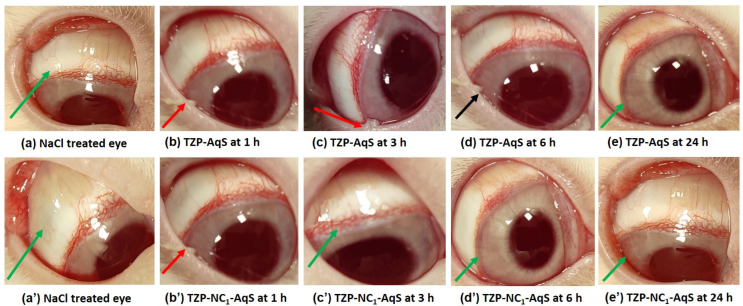
Eye images captured during irritation study. Representative images of 0.9% NaCl-treated eyes (**a**) and (**a’**). Post topical application of conventional TZP-AqS at 1 h (**b**) (red arrow); at 3 h (**c**) (red arrow); at 6 h (**d**) (black arrow); and at 24 h (**e**) (green arrow). Post application of suspension of NC_1_ at 1 h (**b’**) (red arrow); at 3 h (**c’**) (green arrow); at 6 h (**d’**) (green arrow); and at 24 h (**e’**) (green arrow).Images are not showing any abnormal watery discharge or intense redness, indicating the normal features of rabbit eyes, represented by green arrows.

**Figure 3 molecules-27-04619-f003:**
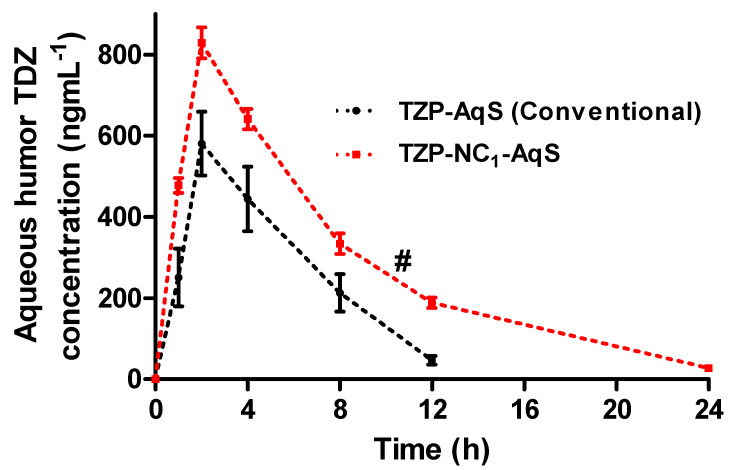
The drug concentration–time profile of AqH samples after topical application of conventional TZP−AqS and TZP-NC_1_−AqS in rabbit eyes. Results are the mean of three measurements (three animals per group) with SEM. ^#^ (*p* < 0.05) represents the significant difference between NC_1_ as compared to conventional AqS.

**Figure 4 molecules-27-04619-f004:**
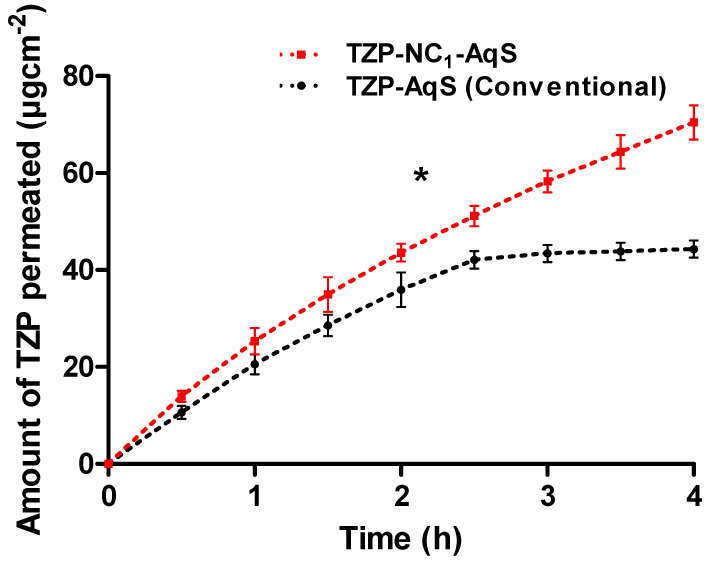
Transcorneal permeation of TZP from conventional TZP−AqSand TZP−NC_1_-AqS (mean ± SD, *n* = 3). * (*p* < 0.05) represents the significant difference between NC_1_ as compared to conventional AqS.

**Table 1 molecules-27-04619-t001:** Composition of TZP-containing formulations.

Ingredients	TZP-NC_1_-AqS (% *w*/*v*)	Conventional TZP-AqS (Prepared in-House) (% *w*/*v*)
Tedizolid Phosphate	0.1	0.1
Ploxamer-188	1.0	-
Benzalkonium chloride	0.01	-
Stearylamine	0.2	-
Mannitol	1.0	-
Polyvinyl alcohol	-	0.5
Dextrose (5%, *w*/*v* solution)	q. s. to 10 mL	q. s. to 10 mL

**Table 2 molecules-27-04619-t002:** Zones of inhibition attained by TZP-NC_1_-AqS and conventional TZP-AqS by agar diffusion test method; the blank-AqS was used as control. Data are the mean of three measurements with SD.

Microorganisms	Diameters of the Zone of Inhibition (mm), Mean ± SD, *n* = 3		
	TPZ-NC_1_-AqS	TPZ-AqS	Blank-AqS
*B. subtilis*	36.43 ± 1.81	28.17 ± 1.32	7.36 ± 0.54
*S. pneumoniae*	37.13 ± 1.93	27.03 ± 1.15	7.53 ± 0.58
*S. aureus*	40.33 ± 1.11	26.35 ± 1.04	7.73 ± 0.46
MRSA (*SA* 6538)	36.77 ± 1.37	25.13 ± 1.28	7.09 ± 0.29
Statistical analysis by one-way ANOVA			
**Tukey’s multiple comparison test**	***p* < 0.05**	**95% CI * of difference**	
TPZ-NC_1_ vs. TZP-AqS	Yes	8.469 to 13.53	
TPZ-NC_1_ vs. Blank-AqS	Yes	27.70 to 32.77	
TZP-AqS vs. Blank-AqS	Yes	16.70 to 21.77	

* CI = Confidence interval.

**Table 3 molecules-27-04619-t003:** Weighted scores for the eye irritation test of TZP-NC_1_-AqS as compared to conational TZP-AqS.

Lesions in the Treated Eyes	Individual Scores for Eye Irritation by					
	TZP-AqS			TZP-NC_1_-AqS		
	In Rabbit			In Rabbit		
	Ist	IInd	IIIrd	Ist	IInd	IIIrd
Cornea						
a. Opacity	0	0	1	0	0	0
b. Involved area of cornea	4	4	4	4	4	4
Total scores = (a × b × 5) =	0	0	20	0	0	0
Iris						
a. Lesion values	0	0	0	0	0	0
Total scores = (a × 5) =	0	0	0	0	0	0
Conjunctiva						
a. Redness	0	0	1	0	1	0
b. Chemosis	0	0	0	0	0	0
c. Mucoidal discharge	0	0	1	0	1	0
Total scores = (a + b + c) × 2 =	0	0	4	0	4	0

**Table 4 molecules-27-04619-t004:** Calculation of maximum mean total score (MMTS) by considering the obtained scores.

TZP-AqS (Conventional)					
In Rabbit	Ist	IInd	IIIrd	SUM	Average (SUM/3)
Cornea	0	0	20	20	6.67
Iris	0	0	0	0	0.00
Conjunctiva	0	0	4	4	1.33
SUM total =	0	0	24	24	8.00
TZP-NC_1_-AqS					
In rabbit	Ist	IInd	IIIrd	SUM	Average (SUM/3)
Cornea	0	0	0	0	0.00
Iris	0	0	0	0	0.00
Conjunctiva	0	4	0	4	4.00
SUM total =	0	4	0	4	1.33

**Table 5 molecules-27-04619-t005:** Ocular pharmacokinetics of TZP-containing formulations. The data are represented as mean with ± SEM of three readings, where ^#^ (*p* < 0.05) represents the significant difference between NC_1_ as compared to conventional AqS.

Parameter	For Conventional TZP-AqS (Mean ± SEM)	For TZP-NC_1_-AqS (Mean ± SEM)	Enhancement Ratios
t_1/2_ (h)	2.66 ± 0.12	4.45 ± 0.18 ^#^	1.67
T_max_ (h)	2.00 ± 0.00	2.00 ± 0.00	Same
C_max_ (ngmL^−1^)	580.92 ± 45.48	829.21 ± 38.27 ^#^	1.43
AUC_0–24h_ (ngmL^−1^h)	3401.68 ± 355.52	6651.25 ± 259.51 ^#^	1.96
AUC_0–∞_ (ngmL^−1^h)	3581.99 ± 382.76	6826.34 ± 256.32 ^#^	1.91
AUMC_0–∞_ (ngmL^−1^h^2^)	18,127.47 ± 2123.36	48,677.57 ± 1697.92 ^#^	2.69
MRT_0–∞_ (h)	5.05 ± 0.054	7.13 ± 0.02 ^#^	1.41
Cl/F (mLh^−1^)	11.43 ± 1.25^#^	5.88 ± 0.22	1.95

**Table 6 molecules-27-04619-t006:** Parameters of transcorneal permeation from conventional TZP−AqS and TZP−NC_1_−AqS (mean ± SD, *n* = 3).

Parameters	TZP-AqS (Conventional)	TZP-NC_1_-AqS
Cumulative amount of drug permeated (µgcm^−2^) at 4th h	44.32 ± 1.74	70.43 ± 3.52
Steady-state flux, *J* (µgcm^−2^h^−1^)	19.18 ± 1.03	31.65 ± 2.39
Permeability coefficient, *P_app_* (cmh^−1^)	(1.92 ± 0.11) × 10^−2^	(3.16 ± 0.24) × 10^−2^
pH	6.18 ± 0.46	7.03 ± 0.35
Corneal hydration level (%)	77.29 ± 1.23	78.05 ± 1.27

## Data Availability

The data presented in this study are available on request from the corresponding author.

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
