# Peer review of "Perspectives of Positively Charged Nanocrystals of Tedizolid Phosphate as a Topical Ocular Application in Rabbits"

_molecules, 2022, doi:10.3390/molecules27144619_

Round 1
Reviewer 1 Report
The article written by Alshememry and coworkers investigates the use of nanocrystal of tedizolid phosphate for topical ocular application to treat bacterial infections. You can find attached my comments. As you can see in my comments, the article requires an extensive improvement in English and the style used.
Moreover, I have a few questions:
1. Which part of the eye do you want to target?
2. Why did you use the Draize test if you can use a less invasive test like the HET-CAM test?

Author Response
Reviewer # 1
Comments and Suggestions for Authors
The article written by Alshememry and coworkers investigates the use of nanocrystal of tedizolid phosphate for topical ocular application to treat bacterial infections. You can find attached my comments. As you can see in my comments the article requires an extensive improvement in English and the style used.
Answer: Thanks for the comments raised on this article. The consideration of the comments and suggestions greatly improved the quality and readability of the article. All the suggestions provided by the reviewer by highlighting the PDF file (peer-review-20363733.v1.pdf) were incorporated into the word file and highlighted in green colour.
Also, the reviewer asked for the name of the bacterial strains under the heading 2.2.3 antimicrobial study. The names of bacterial strains are already mentioned in the next line.
Moreover, I have a few questions:
Query 1. Which part of the eye do you want to target?
Answer: The beauty of nanocrystals is that they can be used for both periocular and intraocular drug delivery. Here we are preparing the nanocrystals for topical use, which can be utilized for the whole eye. Specifically, we are not targeting any particular part of the eye but we are focusing on the bacterial infections associated with the eye.
Query 2. Why did you use Draize’s test if you can use a less invasive test like the HET-CAM test?
Answer: There is no doubt the HET-CAM test is a better alternative to Draize’s test but it is an in-vitro study. We have a good facility for animal use at our premises, with all necessary arrangements. So we opted for Draize’s test rather than the HET-CAM test.
Reviewer 2 Report
The authors attempt to evaluate the safety and efficacy of tedizolid phosphate nanocrystals for medical applications in ophthalmology. However, they state that most of the information about nanocrystallized TZP is unpublished, and so I cannot evaluate the results of this paper based on physical properties. The present study should be submitted only after the unpublished data is made accessible to reviewers in some form.
In addition, it is unclear why low molecular weight TZP, which is a powder at room temperature, would be nanocrystallized rather than used with existing colloidal carriers, and it is this comparison that is important.
Furthermore, I presume the authors adopted sterilized samples for each experiment, but guess that TZP-AqS with a particle size of 574.5 nm would lose most of its main content when sterilized through a 0.22 µm filter, would not the authors agree?
Finally, the authors have used abbreviations in the abstract from their first mention, failed to provide explanations of the abbreviations, and misplaced chapter numbers. A thorough proofreading of the entire paper is strongly recommend.
Author Response
Reviewer #2
Comments and Suggestions for Authors
Query 1. The authors attempt to evaluate the safety and efficacy of tedizolid phosphate nanocrystals for medical applications in ophthalmology. However, they state that most of the information about nanocrystallized TZP is unpublished, and so I cannot evaluate the results of this paper based on physical properties. The present study should be submitted only after the unpublished data is made accessible to reviewers in some form.
Answer: The data relating to NCs development and characterization has been published now and can be accessed at https://doi.org/10.3390/pharmaceutics14071328.
Query 2. In addition, it is unclear why low molecular weight TZP, which is a powder at room temperature, would be nanocrystallized rather than used with existing colloidal carriers, and it is this comparison that is important.
Answer: The drug nanocrystals are pure drug particles with reduced particle size and stabilized by excipients such as surfactants or polymers or a combination of both. The nanocrystals provide the advantage of comparatively high drug load than the existing colloidal carriers which further helps in the efficient transport of drugs to the cells. Further, the production of nanocrystals is less cumbersome than the colloidal carriers, as well as the physical and chemical instability-related issues, are least associated with the nanocrystals. Nanocrystals can be utilized for both periocular and intraocular drug delivery. Literature is also available for the preparation of nanocrystals for ocular drug delivery of low molecular weight APIs such as Brinzolamide (MW 383.50) (https://doi.org/10.1016/j.ijpharm.2014.03.048) which is having low molecular weight than Tedizolid phosphate (MW 450.3). These attributes of nanocrystals guided us to prepare nanocrystals of Tedizolid phosphate for ocular drug delivery which is not reported earlier.
Query3. Furthermore, I presume the authors adopted sterilized samples for each experiment, but guess that TZP-AqS with a particle size of 574.5 nm would lose most of its main content when sterilized through a 0.22 µm filter, would not the authors agree?
Answer: The filtration step was performed with the nanocrystals only while TZP-AqS was not terminally sterilized rather it was prepared in the aseptic area using freshly autoclaved Milli-Q water. As per the query, the necessary modifications are included in the procedure and highlighted in green colour.
Query 4. Finally, the authors have used abbreviations in the abstract from their first mention, failed to provide explanations of the abbreviations, and misplaced chapter numbers. A thorough proofreading of the entire paper is strongly recommend.
Answer: The suggested corrections related to abbreviations were done and highlighted in green color texts. The chapter numbers also formatted in proper sequence. As per the suggestion, thorough proofreading of the manuscript done and necessary corrections were incorporated.
Reviewer 3 Report
I have no any critical remarks for this article. Please consider a few minor recommendations.
Do not use abbreviation in Abstract, or introduce it before using.
Please modify your manuscript according to journal template (especially fonts throughout the manuscript and list of references).
Author Response
Reviewer #3
Comments and Suggestions for Authors
I have no any critical remarks for this article. Please consider a few minor recommendations.
Query 1. Do not use abbreviation in Abstract, or introduce it before using.
Answer: The suggestions are implemented in the abstract and highlighted in green colour.
Query 2. Please modify your manuscript according to the journal template (especially fonts throughout the manuscript and a list of references).
Answer: The manuscript has been modified to only one type of font i.e. “Palatino Linotype”.
Reviewer 4 Report
The article deals with an interesting topic whose purpose is to develop a new nanocrystal for antimicrobial potential for use un pharmaceutics.
Please, add some monocrystal characterization, example Scanning electron microscopy to prove the monocrystal obtained.
Regarding the quality of the article, it would be advisable for the language to be corrected by someone experienced because there are phrases that do not sound scientifically sufficient.
The results and discussions part is acceptable, and the results are clearly presented.
Author Response
Reviewer #4
Comments and Suggestions for Authors
The article deals with an interesting topic whose purpose is to develop a new nanocrystal for the antimicrobial potential for use in pharmaceutics.
Query 1. Please, add some monocrystal characterization, for example, scanning electron microscopy to prove the monocrystal obtained.
Answer: We already published the data associated with Tedizolid phosphate nanocrystals preparation and characterization which can be accessed at the following url:
https://doi.org/10.3390/pharmaceutics14071328
Query 2. Regarding the quality of the article, it would be advisable for the language to be corrected by someone experienced because there are phrases that do not sound scientifically sufficient.
Answer: The manuscript has been thoroughly proofread to maintain its scientific readability and necessary changes were incorporated.
Query 3. The results and discussions part is acceptable, and the results are clearly presented.
Answer: Authors are thankful to the reviewer for the positive response.
Reviewer 5 Report
Dear Authors,
Thank you for submitting your work to MDPI Molecules. Please see comments below.
-- Why does the text size / font change throughout the text? It seems as though perhaps copy and paste was used?
-- Line 105: "The unpublished data regarding these experiments have been communicated as a separate article in another journal." Can the authors provide a reference? Apologies if I missed this.
-- Line 123: I think you mean 0.22 um, not 0.22 u.
-- The paper would benefit from a chart or diagram that visually explains the various animal groups, treatments, tests, etc.
-- The TZP NC synthesis section 2.2.1 lists size, polydispersity index, zeta potential, SEM, FTIR, DSC, XRD, etc. but does not report this information in this paper because it was 'reported in a different paper', but that paper is not cited? It would be helpful to include at least some of that data in this paper, as it is easier for the reader when the results of a paper is complete / self-contained. However, be sure to avoid copyright infringement of course.
-- Is the ocular study at least single blind, no pun intended? As in, does the person scoring the rabbit's eyes have information about the treatment protocol? The best study would ensure that the person scoring the eye condition is COMPLETELY UNAWARE of what treatment the rabbit has been given, including control animals.
-- Can you add some schematic or photograph of the permeation experiments or permeation measurement apparatus? That may be helpful to readers.
Author Response
Reviewer #5
Comments and Suggestions for Authors
Query 1. Why does the text size/font change throughout the text? It seems as though perhaps copy and paste were used?
Answer: We prepared the whole manuscript using “Arial” font with a “12” font size and uploaded the same. The font and text-related changes might have occurred during the conversion of the manuscript to the MDPI template. Truly there is no copy-paste used rather the text was typed and formatted with a single font style.
Query 2. Line 105: "The unpublished data regarding these experiments have been communicated as a separate article in another journal." Can the authors provide a reference? Apologies if I missed this.
Answer: The data relating to the preparation and characterization of Tedizolid phosphate nanocrystals is published now and the reference for the same is incorporated in the manuscript. The reference can be accessed at https://doi.org/10.3390/pharmaceutics14071328
Query 3. Line 123: I think you mean 0.22 um, not 0.22 u.
Answer: The text “0.22µ” has been corrected as “0.22µm” and highlighted in green colour in the manuscript.
Query 4. The paper would benefit from a chart or diagram that visually explains the various animal groups, treatments, tests, etc.
Answer: No doubt this is good suggestion by the reviewer. This could be beneficial when multiple number of groups would be involved in the experiments. In the present investigation we have mainly utilized only two groups of animals, one for developed formulation and one for control formulation. These two groups are well explained in the texts.
Query 5. The TZP NC synthesis section 2.2.1 lists size, polydispersity index, zeta potential, SEM, FTIR, DSC, XRD, etc. but does not report this information in this paper because it was 'reported in a different paper', but that paper is not cited? It would be helpful to include at least some of that data in this paper, as it is easier for the reader when the results of a paper are complete/self-contained. However, be sure to avoid copyright infringement of course.
Answer: As we mentioned in the reply to the second query, the data has been published. Now it will be difficult to incorporate the same data without permission from the publisher. For the ease of readers, the reference has been included in the manuscript.
Query 6. Is the ocular study at least single-blind, no pun intended? As in, does the person scoring the rabbit's eyes have information about the treatment protocol? The best study would ensure that the person scoring the eye condition is COMPLETELY UNAWARE of what treatment the rabbit has been given, including control animals.
Answer: We thanks the reviewer for such a nice suggestion, in the present investigation the scoring for eye irritation/conditions was performed by a person who was completely unaware of what treatment the rabbits were given during the irritation study.
Query 7. Can you add some schematic or photograph of the permeation experiments or permeation measurement apparatus? That may be helpful to readers.
Answer: As per the suggestion, the schematic of the Franz Diffusion Cell has been created. The schematic representation will be incorporated in the supplementary file as shown below
Figure S1: Schematic representation of Franz Diffusion cell
Also attached as PDF

Round 2
Reviewer 2 Report
I have no complaints about the corrections and responses to my comments.